# Needs of Children with Neurodevelopmental Disorders and Geographic Location of Emergency Shelters Suitable for Vulnerable People during a Tsunami

**DOI:** 10.3390/ijerph18041845

**Published:** 2021-02-14

**Authors:** Hisao Nakai, Tomoya Itatani, Seiji Kaganoi, Aya Okamura, Ryo Horiike, Masao Yamasaki

**Affiliations:** 1School of Nursing, Kanazawa Medical University, 1-1 Uchinada, Kahoku, Ishikawa 920-0265, Japan; h-nakai@kanazawa-med.ac.jp; 2School of Health Sciences, College of Medical, Pharmaceutical and Health Sciences, Kanazawa University, 5-11-80 Kodatsuno, Kanazawa, Ishikawa 920-0942, Japan; 3Department of Rehabilitation, Geisei Hospital, 4268 Wajiki, Geisei, Aki, Kochi 781-5701, Japan; kaganois@mizukikai.or.jp (S.K.); geisei-reha@mizukikai.or.jp (A.O.); 4Susaki Regional Welfare and Health Center, 6-26 Higashifuruichimachi, Susaki, Kochi 785-8585, Japan; spi4949@gmail.com; 5Kochi Mental Health and Welfare Center, 2-4-1 Marunouchi, Kochi 780-0850, Japan; masao_yamasaki@ken2.pref.kochi.lg.jp

**Keywords:** children with neurodevelopmental disorders, disaster preparedness, emergency shelter, evacuation, tsunami

## Abstract

In the current study, we sought to identify special needs and safe evacuation conditions for children with neurodevelopmental disorders (CNDs) along Japan’s tsunami-prone Pacific coast. A survey and spatial analysis were used to collect data of CNDs (*n* = 47) and their caregivers. Areas predicted to be flooded in a tsunami, as well as evacuation routes to emergency shelters for vulnerable people (ESVPs), were mapped using geographic information systems (GIS). Our results showed that five professional staff were needed to support 33 CNDs requiring 135.9 m^2^ of ESVP space. Critical safety factors were altitude, vertical evacuation, accessibility to ESVPs, and nonexistence of estuaries in the direction of evacuation. GIS-based spatial analysis and evacuation modeling for disaster preparedness and training plans that involve nurses are essential.

## 1. Introduction

Since the Great East Japan Earthquake of 2011, natural disasters such as earthquakes and heavy rain and snowfall have occurred frequently in Japan [1]. Tsunamis, caused by earthquakes, have a particularly large human impact. Tsunami disaster countermeasures are an important issue for cities on the Pacific Ocean side of Japan. These cities are vulnerable to an earthquake occurring in the Nankai Trough, which is expected to occur in the near future. Studies have predicted that the magnitude of the earthquake will be similar to that of the 1845 Ansei Nankai earthquake [2]. The resulting tsunami is expected to cause numerous human casualties in coastal cities [3]. The height of the tsunami caused by the Great East Japan Earthquake reached over 9.3 m [4] and resulted in 22,000 dead and missing people [5]. The victims were more likely to be older people because it took time to evacuate those in hospitals or nursing homes or those living alone [6]. The Japanese government considers that older people, those with intractable diseases, disabled people, pregnant women, and children with neurodevelopmental disorders (CNDs), require special assistance during an evacuation [7]. Facilities for older and disabled people are designated as “emergency shelters for vulnerable people”, and these are designed to accommodate those who require special assistance during an evacuation. These evacuation shelters are opened when necessary [8]. CNDs may develop variable sensitivities or hypo-responsiveness to various stimuli. Specific symptoms may be affected by a variety of sensory patterns, such as responsiveness to loud sounds, dislike of water, hypersensitive taste, lack of attention, ignoring loud sounds, and not responding to names. Behaviors and symptoms caused by abnormal reactivity include aversion to light, covering the ears, and avoiding contact with the skin [9]. Rogers, Hepburn, and Wehner showed that symptoms could include abnormal sensory responsiveness, such as abnormal taste, smell, and auditory responsiveness, and hypo-responsiveness to stimuli [10]. Stough found that emergency sirens could stimulate and stress these children [11]. During the 1995 Great Hanshin-Awaji Earthquake, parents of CNDs often chose to keep them in cars or away from shelters to avoid disturbing others [12]. CNDs may exhibit problematic behaviors such as aggression, hyperactivity, and anxiety following unexpected changes in their schedule and stress owing to environmental changes [13,14]. However, the symptoms of CNDs are difficult for others to understand. In the 2016 Kumamoto Earthquake, some CNDs were not allowed to stay in evacuation shelters because their unbalanced eating and panic were perceived as “selfish” [15].

For CNDs to evacuate quickly in the event of a disaster, it is necessary to know the location, route, and geographical characteristics of the nearest shelter and prepare for evacuation. Network analysis using geographic information systems (GISs) is suitable for determining the reachable range from the optimum shelter, route, and travel speed. Thus far, there have been few studies using GIS to help vulnerable people evacuate in the event of a tsunami. Analysis of shelter selection based on hurricane strength and shelter demand has been conducted using the interdiction and median model [16]. Evaluation of existing shelters using the *p*-median problem has been reported in a part of the target area (Aki City) of this study [17]. Estimations of human harm has been carried out using buffer analysis of a tsunami vertical shelter [18]. Evacuation time evaluation [19] is conducted to identify the population at risk of injury in a tsunami and considers the number of evacuation vehicles and congestion. Establishment of effective signposting to help the hearing impaired to evacuate [20] has been reported.

Reports on evacuation from a tsunami and evaluation of shelters have used sensitivity analysis with least-cost distance (LCD) modeling [21,22]. Evacuation routes and shelter allocation methods to minimize casualties [23] as well as evaluation of the effectiveness of protection in vertical evacuation from a near-field tsunami [24], have been reported. On reports targeting vulnerable people, Emergency Evacuation Readiness of Full-Time Wheelchair Users with Spinal Cord Injury [25], Simulation for vulnerable people to evacuate by car [26], Simulation of tsunami evacuation guidance signs for the hearing impaired [20], evacuation simulation of people using medical devices from a tsunami [27]. However, no studies have focused on local CNDs, estimated tsunami damage and evacuation times, or the geographic requirements to protect these vulnerable children.

The aim of this study was to clarify the needs of CNDs who live in the coastal areas of Konan, Geisei, and Aki in Kochi Prefecture, Japan and to calculate the required amount of personal space and professional support needed by this population. Using GIS, we also identified evacuation routes from children’s homes to the nearest emergency shelter for vulnerable people (ESVP), and identified the geographical conditions required for ESVPs, to protect these children during a tsunami. Analysis using the actual place of residence of CNDs and the location of existing ESVPs will be useful in developing evacuation measures tailored to the actual situation of CNDs. Clarifying the disaster needs of CNDs will help municipal managers to plan the installation of appropriate shelters.

## 2. Materials and Methods

### 2.1. Terminology

#### Emergency Shelter for Vulnerable People (ESVP)

An ESVP is a facility specially prepared for people who have difficulty staying in general shelters, and their families. The Basic Law on Disaster Management defines an ESVP as a shelter for elderly people, pregnant women, people with illnesses and disabilities, and people with developmental disorders and intractable diseases. Kochi Prefecture has designated the existing welfare facility as an ESVP. The decision to open an ESVP in the event of a disaster is made by the head of the local government. If there are vulnerable people in a general shelter, the head of the local government will request the facility manager to open the ESVPs [7]. Most municipalities do not cover the costs of opening and maintaining ESVPs in the event of a disaster.

### 2.2. Data Collection

We conducted a survey with the cooperation of a hospital in the village of Geisei, Kochi Prefecture that treats CNDs who require special assistance. Two physiotherapists, who are in charge of functional training for CNDs, interviewed 47 children and their caregivers. The physiotherapists collected data using the Kanazawa and Kochi Disaster Preparedness System (K-DiPS). This is a system that uses an electronic device to understand the needs for medical treatment and other attention in the event of a disaster. The K-DiPS has a built-in checklist to collect information to help with the evacuation of disabled people. People in need of support, their caregivers, and health care professionals complete the K-DiPS checklist together. In this survey, we did not use electronic devices but rather we conducted interviews using a printed version of the checklist. The information collected on the K-DiPS checklist includes sex, date of birth, address, primary condition, medical treatment, medication, intention to evacuate, difficulty with activities of daily living, assistance required in the event of a disaster, and personal space requirements. The physiotherapists were shown in advance how to use the K-DiPS checklist and were trained in interviews with simulated patients. Data were collected between January 2018 and March 2019.

### 2.3. Target Area

The target areas were the cities of Konan and Aki and the village of Geisei, in the Shikoku region. These three areas are all located on the Pacific coast and are adjacent to each other (Figure 1). The population is about 32,000 in Konan, 3000 in Geisei, and 17,000 in Aki (Figure 2) [28].

An earthquake along the Nankai Trough is expected to occur within the next 30 years, and tsunamis are predicted to occur along the Pacific coast as a result [29]. A 30-cm tsunami is expected to reach Aki first, within 5 min of the earthquake. It is expected to arrive in Geisei within 5 to 10 min and to reach Konan within 10 to 20 min [30]. 

### 2.4. Overview of Participants and Their Needs

The attributes and main conditions of the 47 CNDs surveyed in this study are shown in Table 1. 

According to the K-DiPS checklist completed for the 47 CNDs included in the study, participants had the following characteristics: “Hyperactive and may go anywhere if left unattended”, 26 CNDs (55.3%); “Makes loud voices wherever you are owing to poor resistance to frustration”, 21 (44.7%); and “Extremely unbalanced diet”, 21 (44.7%); among others. The items regarding the need for help in the event of a disaster were as follows: “Hyperactive and may go anywhere if left unattended” 24 (51.1%), “Makes loud voices wherever you are owing to poor resistance to frustration” 21 (44.7%), “Extremely unbalanced diet” 19 (40.4%) children.

### 2.5. Mapping Locations of CNDs, ESVPs, and Areas Predicted to Be Flooded in a Tsunami

We obtained the address of ESVPs in Konan, Geisei, and Aki from information published on the Internet. Using GIS, we converted these addresses and the addresses of CNDs into latitude and longitude. On all maps, locations of the homes of CNDs are shown as green squares and the emergency shelters as blue circles. We obtained information on the estimated tsunami inundation area, sediment disaster hazard areas, and rivers from a government website [31], which details the inundation area and depth for the largest class tsunami [30]. We color-coded the inundation area according to inundation depth on the maps using ESRI ArcGIS Pro 2.3.4.

### 2.6. Space and Professional Support Needs of Children with Neurodevelopmental Disorders (CNDs)

We explored the average age, sex, primary condition, specific difficulties, and need for space among CNDs. We based our standards for setting up a children’s developmental support center on those established by the Ministry of Health, Labor and Welfare and set the area of intrinsic space required for children in need of evacuation to 4.12 m^2^/person [32]. The number of professionals required to support the children was set at two or more for up to 10 evacuated children and one for every one to five additional children [33]. We then calculated the specific space and number of professional support staff required for CNDs to live in ESVPs. Aki City has established the number of people that can be accommodated in each facility, so we calculated the difference between the number of people who can stay for more than a week at facilities in Aki City and the number of CNDs.

### 2.7. Analysis of the Geographic Location of ESVPs

We counted the number of CNDs and the ESVPs located in the predicted tsunami inundation area. Using Network Analyst, we calculated the range that a child could be evacuated on foot in the time between an earthquake and the arrival of a tsunami. Network Analyst is a function of ArcGIS that draws a movable range based on the conditions of the roads that may be used, moving speed, and time. The cutoff range divides the movable range by time and is shown in different colors on a map. Based on the results of the Great East Japan Earthquake, we assumed that walking speed would be 2.24 km/h [34]. We also analyzed the route from children’s homes to the nearest ESVP. We determined the route with least risk in consideration of the road damage caused by an earthquake and the effects of flooding caused by a tsunami. Specifically, among the national roads and prefectural roads, locations designated by the national government as at risk of becoming impassable included sediment-related disaster hazard areas, bridges over the Koso, Karasu, Yasu, Wajiki, Akano, Aki, and Ioki rivers designated as Class B rivers, and bridges in the tsunami inundation area. We conducted route analysis based on these parameters.

## 3. Results

### 3.1. Overview of ESVPs

In the target area, 17 facilities have been designated as ESVPs and have been assigned labels from A to Q. Of these, 5 (29.4%) ESPVs were located in the inundation prediction area; specifically, these were A, F, H, P, and Q. Information regarding inundation depth, straight-line distance from ESVP to the coast, CNDs expected to evacuate, and municipality is shown in Table 1.

### 3.2. Space and Professional Support Required 

In total, 33 (70.2%) respondents had a need for personal space, totaling 135.9 m^2^. The Ministry of Health, Labor and Welfare standards for the establishment of a child developmental support center require the ability to support 24 children who are “hyperactive and may go anywhere if left unattended.” We calculated that five professional supporters were needed for that number of children, and 10 if all 47 participants were evacuated. Aki City ESVPs were calculated to have a total capacity of 60 people. However, the number of evacuees to ESVPs O and Q was each one more than the permitted number.

### 3.3. Location of ESVPs and Evacuation Routes for Children

Figure 3, Figure 4 and Figure 5 are maps for each area showing the predicted depth of inundation and the location of children, as well as the Pacific Ocean to the south. A tsunami would flow from south to north and up rivers from the Pacific Ocean. In total, 15 CNDs lived in the predicted inundation area, including 12 in Aki, two in Geisei, and one in Konan. ESVPs A, F, H, P, and Q were also located in the predicted inundation area. Table 2 shows the estimated inundation depth, the straight-line distance from the emergency shelter to the coast, and the ESVPs located in areas expected to be inundated. No children were expected to evacuate to sites A and H. The nearest ESVP for participant ID42 was F, which would require them to move toward the tsunami. IDs 8, 12, 13, 16, 17, 20, 21, 23, 45, and 47 were predicted to evacuate to ESVP Q, and IDs 13, 17, 20, 24, and 45 would reach the shelter within 5 min. IDs 23 and 47 would move from outside to inside the flooded area. IDs 20, 21, 24, and 45 would move toward the tsunami because they lived further from the coast than ESVP Q. IDs 8, 12, 16, and 17 lived closer to the coast than Q, and ID 17 could reach the facility within 10 min. IDs 8 and 16 would probably not reach Q within 10 min if they traveled through the predicted inundation area. However, they could evacuate within 5 min if they moved straight to the nearest mountainside. IDs 2, 25, 33, and 36 would be unable to cross the bridge over the Ioki River, so they would be unable to evacuate to the ESVP (Figure 3). There are two evacuation centers for Geisei, with J outside the inundation area on the border with Aki, and K in Konan City. IDs 27 and 37 lived in the inundation area and could leave the inundation area within 5 min if moving toward J. ID7 lived outside the inundation area, about 400 m from the coast, and thus could remain at home. However, the estimates suggested that the surrounding area would be flooded, isolating the house. If ID7 moved toward J, evacuation would be possible within 10 min via an area with a water depth of less than 5–10 m. When moving toward J, ID39 was estimated to reach an inundation area with a water depth less than 3–5 m within about 10 min (Figure 4).

## 4. Discussion

### 4.1. Specific Difficulties of CNDS and Support Needed during a Disaster

The present results suggest that there would be a shortage of evacuation space and of specialized support for CNDs at ESVPs in Aki during a disaster. During the 1995 Great Hanshin-Awaji Earthquake, parents of CNDs often chose to keep them in cars or away from shelters to avoid disturbing others [12]. CNDs may exhibit problematic behaviors such as aggression, hyperactivity, and anxiety following unexpected changes in their schedule and stress owing to environmental changes [13,14]. These stress reactions are difficult for others to understand and are particularly easy to overlook in a disaster emergency. Approximately 70% of CNDs in our study needed individual space, and about half were described as “hyperactive and may go anywhere if left unattended.” Emergency evacuation shelters should be equipped with professional supporters and sufficient space to manage these characteristics in CNDs.

### 4.2. Conditions in ESVPs for CNDs

The best evacuation route from a tsunami saves the maximum number of lives in the shortest time [35]. The route should consider the runup in rivers and landslides owing to a tsunami caused by an earthquake. However, if the shortest route traverses lowlands or an area predicted to flood, the risk to safety may be increased. Optimal evacuation routes from tsunamis should therefore also consider the altitude. In our study, even if ID42 started travelling immediately to ESVP F, they would not arrive within 10 min and would move toward the tsunami inflow. It is also unlikely that they could leave home immediately. The effectiveness of vertical evacuation has been pointed out in evacuation from a near-field tsunami [24]. In Japan, it has been reported that tsunami evacuation towers for vertical evacuation may move from highlands to lowland tsunami evacuation towers, increasing the risk of damage [36]. The location of evacuation shelter F, on the coast, may encourage people to move toward the tsunami. Evacuation centers Q and P are also considered to have these risks. IDs 23 and 47 were both closest to evacuation shelter Q, but reaching the shelter would require them to move toward the tsunami from outside the area predicted to be inundated, increasing the risk to their safety. Evacuation shelter P is located in a riverside area near the estuary of Akigawa, with a straight-line distance of about 400 m from the coast. It is therefore expected to receive tsunami inflows from both the south (the sea) and west (the river), and the grace period for reaching the shelter before the surrounding area floods may be shorter than expected. It is therefore recommended that evacuation to F, Q, and P should be limited to those who can reach the higher floors of the building before the expected tsunami arrival time (people who can evacuate vertically). Generally, in areas at high risk of a tsunami, a strong, tall building is recommended as a vertical evacuation site [37,38], however, instead of blindly heading to such a building, the location and tsunami flow in the immediate area should be considered, as well as the direction of arrival. Both horizontal and vertical evacuation should be considered for areas with a short time to tsunami arrival, such as cities in flat areas [18,35]. F, P, and Q are expected to be used by both children requiring assistance and residents along the coast who need to be evacuated vertically. It is therefore necessary to inform the residents of this and to begin stockpiling and secure sufficient power for the number of people expected to evacuate.

### 4.3. Importance of Planning Disaster Preparedness by Nurses and Public Health Nurses Using GIS

Uniform measures are not sufficient for children and other people with neurodevelopmental disorders in the community, even those with the same condition, because of the effect of factors such as age and complications. CNDs have symptoms such as aggression, hyperactivity, and anxiety, in addition to their core symptoms [14]. Substantial environmental changes and unfamiliar stimuli may exacerbate their symptoms. Children who require special health care are particularly vulnerable to disasters, and preparedness based on their inherent health issues and special training for emergencies is needed [39,40]. Coordination with local service providers can increase preparedness and training for managing emergencies [41]. It is therefore important for nurses and public health nurses in charge of support to work with children and families who would need to evacuate. To safely evacuate from a tsunami [42], it is necessary to understand the time until the tsunami will arrive and assume that residents along the coast will evacuate vertically [43].

GIS can be used to visualize all information in space and perform simulation and spatial analysis [44]. Evacuation shelters and routes can be described on a map, and time and distance can be calculated, making it easy to visually check for possible flooding, landslides, and risks to those in need of attention. In the field of public health, many measures to ensure an appropriate public health response in the event of a disaster using GIS [45,46] and prompt needs assessment [47] have been reported. However, there are no reports on the knowledge and training of health professionals, such as public health nurses, needed to put GIS into practice. In Japan, some universities use GIS to educate public health nurses in community diagnosis [48,49], but GIS is used exclusively by health and community health care scholars and not health care practitioners.

Visiting nurses and public health nurses can contribute to the planning of individual evacuation plans and training plans, using GIS to increase awareness about the needs of vulnerable children, their families, and other staff.

## 5. Limitations

The target area and route analysis did not consider the effect of road destruction from earthquakes, traffic congestion, and weather. The cut-off range in the target area analysis did not consider the time required to prepare to leave for a shelter. The study area was limited to Konan, Geisei, and Aki, and ignored other ESVPs outside these three municipalities. To increase the validity of the study, it is necessary to expand the target area, and add preparation time. In cooperation with the hospital associated with the physiotherapists who collected the information, we have begun to review measures for disaster-sensitive children in the target area using the results of this study.

## 6. Conclusions

ESVPs should conform to the following conditions: (1)ESVPs should be located at an altitude that a tsunami cannot reach.(2)No matter what route evacuees choose, they should be able to move from lower to higher ground and avoid flooding, even if they cannot reach a shelter.(3)Emergency evacuation sites in flooded areas must be high-rise buildings that allow for vertical evacuation.(4)Evacuees should all be able to reach an evacuation site before the tsunami arrives.(5)The shelter should be reachable even with tsunami inflow from rivers.(6)There should be no estuaries in the direction of evacuation.

Modeling evacuation for vulnerable people in the event of a disaster and involving nurses or public health nurses in the use of GIS may contribute to better disaster preparedness and training plans.

## Figures and Tables

**Figure 1 ijerph-18-01845-f001:**
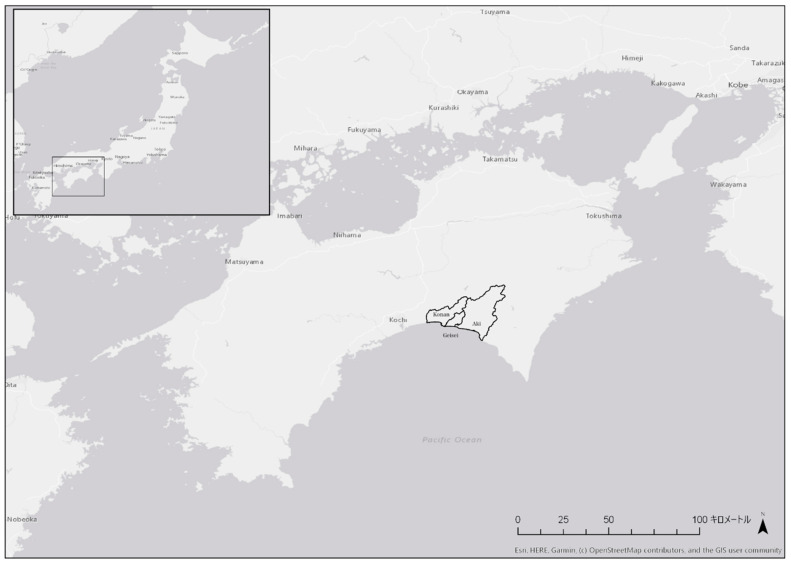
Locations of the cities of Konan and Aki, and the village of Geisei in the Shikoku region of Japan. キロメートル: km.

**Figure 2 ijerph-18-01845-f002:**
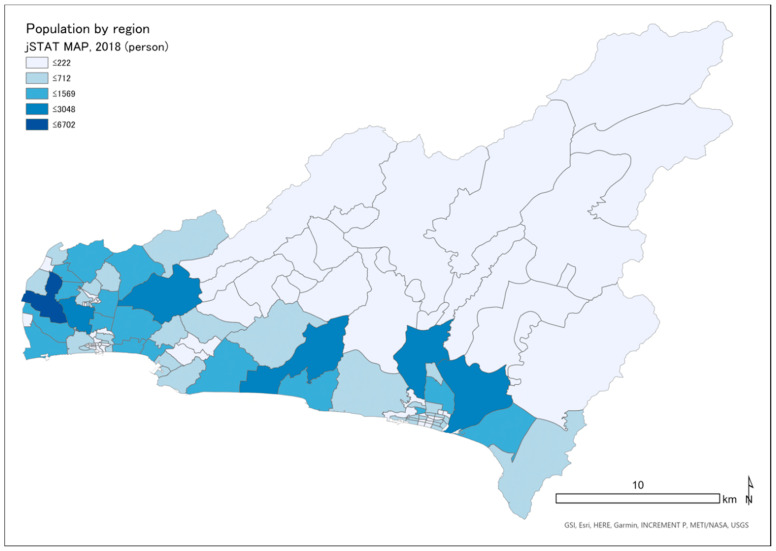
Population distribution by region in Konan, Geisei, and Aki.

**Figure 3 ijerph-18-01845-f003:**
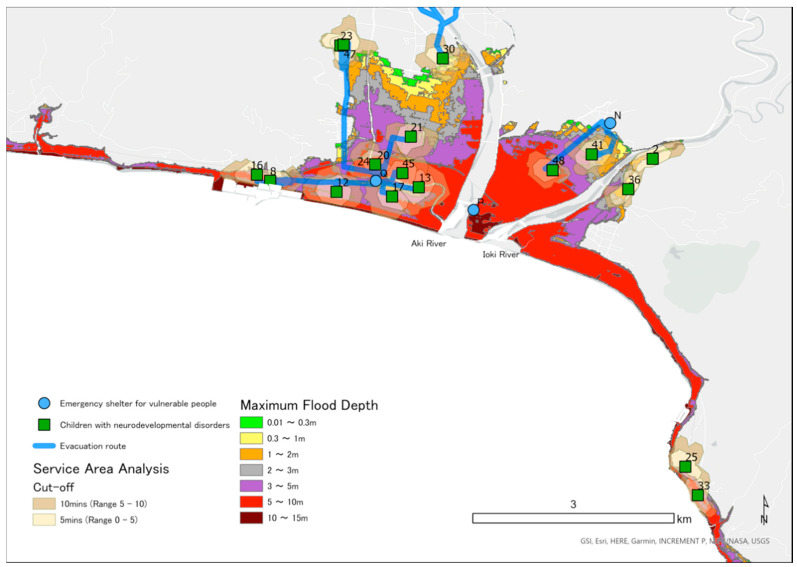
Service area analysis of emergency shelters for vulnerable people (ESVPs) in Konan and evacuation routes to nearby ESVPs.

**Figure 4 ijerph-18-01845-f004:**
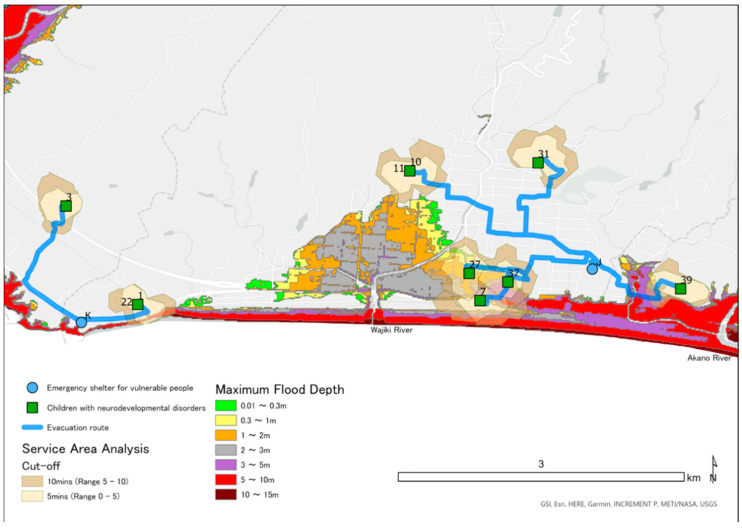
Service area analysis of emergency shelters for vulnerable people (ESVPs) in Geisei and evacuation routes to nearby ESVPs.

**Figure 5 ijerph-18-01845-f005:**
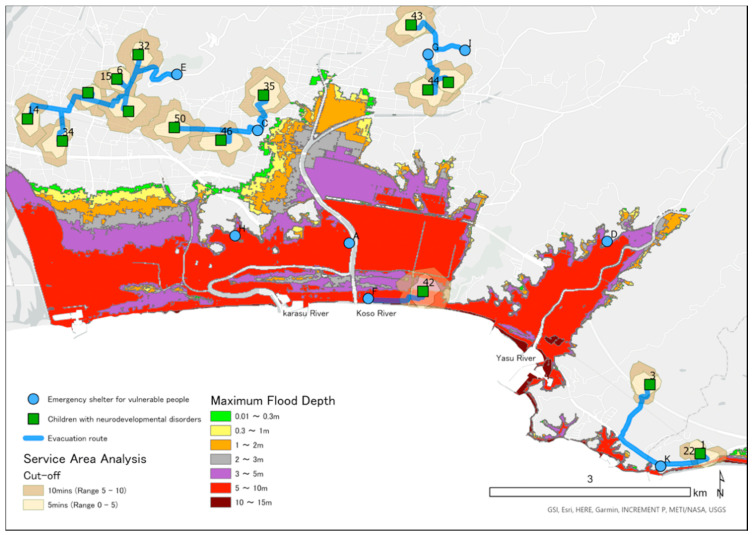
Service area analysis of emergency shelters for vulnerable people (ESVPs) in Aki and evacuation routes to nearby ESVPs.

**Table 1 ijerph-18-01845-t001:** Participants’ attributes and primary condition (*n* = 47).

Item	Category	*n* (%)
Age	Average ± standard deviation	5.7 ± 2.8
Gender	Male	38 (80.9)
	Female	9 (19.1)
Main condition	Autism spectrum disorder	20 (42.6)
	Developmental coordination disorder	17 (36.2)
	Attention deficit hyperactivity disorder	7 (14.9)
	Other	3 (6.4)

**Table 2 ijerph-18-01845-t002:** Characteristics of ESVPs in the predicted inundation area.

ESVP Located in the Inundation Prediction Area	Inundation Depth (m)	Straight Line Distance from ESVP to Coast (m)	CNDs Expected to Evacuate	Municipality
A	5–10	866.4	-	Konan City
F	5–10	77.5	ID42	Konan City
H	2–3	1643	-	Konan City
P	5–10	416.9	ID25, 33	Aki City
Q	3–5	450.4	ID8, 12, 13, 16, 17, 20, 21, 23, 45, 47	Aki City

Abbreviations: ESVP, emergency shelter for vulnerable people; CND, children with neurodevelopmental disorders.

## Data Availability

Data available on request due to restrictions privacy. The data presented in this study are available on request from the corresponding author. The data are not publicly available due to protection for personal data.

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
