# Peer review of "Needs of Children with Neurodevelopmental Disorders and Geographic Location of Emergency Shelters Suitable for Vulnerable People during a Tsunami"

_ijerph, 2021, doi:10.3390/ijerph18041845_

Round 1

Reviewer 1 Report

This paper develops an approach to assist the needs of children with disabilities living in coastal communities supported by a GIS-based analysis to identify the evacuation routes needed to safely transport them to the closes shelter, and the conditions these shelters should be in to serve them.

First of all, there is a definite need to broaden the literature review with papers that focus on the use of GIS for identifying the closest shelter for vulnerable populations such as seniors and people with disabilities. In addition, there are studies that use p-median and r-interdiction models to identify the best shelter or repurpose another shelter to serve these vulnerable segments of the population. Without a solid discussion based on the existing literature, the authors cannot possibly explain what novelty their approach has over the other models in the literature. I would suggest the authors review the works of prominent researchers such as Mark Horner.

One angle I specifically liked was the suggestions on training the health professionals with knowledge on GIS. Are there other attempts in the literature that tried to achieve this? What are the advantages of the proposed model over others? Please elaborate on this issue, it is critical.

It would also be relevant to have a more detailed section on the study area and the roadway network and demographics, possibly with multiple maps.

There should be more sensitivity analyses based on different types of disruptions in order to see how the GIS model reacts to changes in its’ input parameters. The authors consider storm surge maps; however, do not show how the roadways themselves are affected by storm surge and flooding. If an evacuation route is flooded, is there any other route that should be considered? For this type of analysis, “closeness” should not be the only factor considered. There should be a major of risk for different routes, and it may be wise to select the one with least risk. The authors have to address this critical comment entirely.

Author Response

Dear Reviewer: 1

First of all, there is a definite need to broaden the literature review with papers that focus on the use of GIS for identifying the closest shelter for vulnerable populations such as seniors and people with disabilities. In addition, there are studies that use p-median and r-interdictioAn models to identify the best shelter or repurpose another shelter to serve these vulnerable segments of the population. Without a solid discussion based on the existing literature, the authors cannot possibly explain what novelty their approach has over the other models in the literature. I would suggest the authors review the works of prominent researchers such as Mark Horner.

P2 L60-83

Introduction

I modified the document as follows:

For CNDs to evacuate quickly in the event of a disaster, it is necessary to know the location, route, and geographical characteristics of the nearest shelter and prepare for evacuation. Network analysis using geographic information systems (GISs) is suitable for determining the reachable range from the optimum shelter, route, and travel speed. Thus far, there have been few studies using GIS to help vulnerable people evacuate in the event of a tsunami. Analysis of shelter selection based on hurricane strength and shelter demand has been conducted using the interdiction and median model [16]. Evaluation of existing shelters using the p-median problem has been reported in a part of the target area (Aki City) of this study [17]. Estimations of human harm has been carried out using buffer analysis of a tsunami vertical shelter [18]. Evacuation time evaluation [19] is conducted to identify the population at risk of injury in a tsunami and considers the number of evacuation vehicles and congestion. Establishment of effective signposting to help the hearing impaired to evacuate [20] has been reported.

Reports on evacuation from a tsunami and evaluation of shelters have used sensitivity analysis with least-cost distance (LCD) modeling [21, 22], Evacuation routes and shelter allocation methods to minimize casualties [23] as well as evaluation of the effectiveness of protection in vertical evacuation from a near-field tsunami [24], have been reported. On reports targeting vulnerable people, Emergency Evacuation Readiness of Full-Time Wheelchair Users with Spinal Cord Injury [25], Simulation for vulnerable people to evacuate by car [26], Simulation of tsunami evacuation guidance signs for the hearing impaired [20], evacuation simulation of people using medical devices from a tsunami [27]. However, no studies have focused on local CNDs, estimated tsunami damage and evacuation times, or the geographic requirements to protect these vulnerable children.

One angle I specifically liked was the suggestions on training the health professionals with knowledge on GIS. Are there other attempts in the literature that tried to achieve this?

P10 L310-316

I modified the document as follows:

In the field of public health, many measures to ensure an appropriate public health response in the event of a disaster using GIS [46, 47] and prompt needs assessment [48] have been reported. However, there are no reports on the knowledge and training of health professionals, such as public health nurses, needed to put GIS into practice. In Japan, some universities use GIS to educate public health nurses in community diagnosis [49, 50], but GIS is used exclusively by health and community health care scholars and not health care practitioners.

What are the advantages of the proposed model over others? Please elaborate on this issue, it is critical.

P2 L89-92

I modified the document as follows:

Analysis using the actual place of residence of CNDs and the location of existing ESVPs will be useful in developing evacuation measures tailored to the actual situation of CNDs. Clarifying the disaster needs of CNDs will help municipal managers to plan the installation of appropriate shelters.

It would also be relevant to have a more detailed section on the study area and the roadway network and demographics, possibly with multiple maps.

There should be more sensitivity analyses based on different types of disruptions in order to see how the GIS model reacts to changes in its’ input parameters. The authors consider storm surge maps; however, do not show how the roadways themselves are affected by storm surge and flooding. If an evacuation route is flooded, is there any other route that should be considered? For this type of analysis, “closeness” should not be the only factor considered. There should be a major of risk for different routes, and it may be wise to select the one with least risk. The authors have to address this critical comment entirely.

Figure 2

I have color-coded the population distribution by district.

I did an analysis considering the Sediment Disaster Hazard Area and rivers and have added the following text:

P6 L177-184

Analysis of the geographic location of ESVPs

We determined the route with least risk in consideration of the road damage caused by an earthquake and the effects of flooding caused by a tsunami. Specifically, among the national roads and prefectural roads, locations designated by the national government as at risk of becoming impassable included sediment-related disaster hazard areas, bridges over the Koso, Karasu, Yasu, Wajiki, Akano, Aki, and Ioki rivers designated as Class B rivers, and bridges in the tsunami inundation area. We conducted route analysis based on these parameters.

I also added the following text:           

P5 L153-155

Mapping of location of CND, ESVP, and the area predicted to be inundated by a tsunami

We obtained information on the estimated tsunami inundation area, sediment disaster hazard areas, and rivers from a government website [31],

P7 L227-228

Results

IDs 2, 25, 33, and 36 would be unable to cross the bridge over the Ioki River, so they would be unable to evacuate to the ESVP (Fig. 3).

Reviewer 2 Report

The article analyzes the needs of children with neurodevelopmental disabilities related to tsunami emergencies in a geographical area of ​​Japan threatened by this type of event.

It is a subject with an evident social interest that can have practical implications of great importance.

The methodology used is rigorous and the conclusions are correctly justified from the results.

My main concern is in the use of the term "developmental disabilities."
The authors do not give a clear definition of their use of the term.
Most of the characteristics mentioned in the introduction (sensitivities or hyporesponsiveness to various stimuli, responsiveness to loud sounds, dislike of water, hypersensitive taste, lack of attention, ignoring loud sounds and not responding to names) are common symptoms of autism spectrum disorder (ASD).
But the term "developmental disabilities" is much broader, encompassing diagnoses other than ASD.
My recommendation is to use the label "neurodevelopmental disabilities."

According to the Diagnostic and Statistical Manual of Mental Disorders (DSM-5), this term includes children with ASD, intellectual disabilities, attention deficit hyperactivity disorder (ADHD), and various motor disorders (among others). These diagnoses are precisely those that appear in Table 1.
I think that the introduction should contain a definition of the term "neurodevelopmental disorders" and that this term should be used throughout the article.

Additionally, other symptoms should be added to the list of symptoms to characterize these children, such as cognitive limitations, difficulties in understanding social contexts or in identifying the causes of what is happening around them. These symptoms can be very relevant in natural emergencies.

Here are some other questions that I think the authors should consider:

1. Introduction.
In the introduction practically no previous studies are mentioned that have addressed the same objectives as this article.
The introduction should be expanded by framing this article in the field of research on the location of shelters for natural emergencies for the general population; and for people with disabilities in particular.

2. Emergency shelter for vulnerable people (ESVP).
This article may be of interest to researchers specialized in natural emergency response analysis who are very familiar with these issues.
But it may also be of interest to researchers interested in the analysis of public policies aimed at people with disabilities. In this case, these researchers will hardly have knowledge of natural emergencies and their response, and they need some context to understand the article.
Although in Japan it's probably be obvious, some issues should be explained in the introduction. For example:
Who decides where, when and with what features to build these ESVPs? The local governments, the national government?
Who bears the costs of its construction and maintenance?
Children with neurodevelopmental disorders who must live in these ESVPs for days, are they accompanied by their relatives? Or are they only accompanied by professionals?
This information should be added in lines 65-68.
I suggest authors to put themselves in the shoes of a person who is not at all familiar with earthquake or tsunami alerts, and to explain issues that, although they may seem basic in Japan, are not in many contexts.

3. Structure of the article.
3.1. Table 1.
Table 1 describes the characteristics of the participants. It should be relocated to the Materials and methods section, not the Results section.

3.2. Lines 200-209.
This content is more appropriate for the introduction section, not for the discussion.

3.3. Lines 244-254.
This content is repeated in the Conclusions section. It is suggested to delete these lines and leave their content only in the Conclusions section.

4. Figure 2.
Regarding the format of the figure 2, I consider that it is not necessary to represent this information with a graph. It would be enough to explain it through text.
Regarding its content, have these results been prepared from the results of the K-DiPS?
If so, it should be specified.

5. Lines 150-151.
Before referring to the specific shelters O and Q, the number of existing shelters should be reported and that each one is to be named by a letter.
It is possible that this is a well-known issue in Japan. Most of the potential readers of the article who do not live in areas threatened by tsunamis are not familiar with this.

6. Discussion.
In the discussion, the results of the present study should be compared with those of other possible previous studies that have worked on this same issue.

Formal issues:
1. Keywords must be in alphabetic order.
2. It must be indicated the meaning of "GIS" the first time it appears (it is currently explained on line 100).

Author Response

Dear Reviewer: 2

My main concern is in the use of the term “developmental disabilities.”

The authors do not give a clear definition of their use of the term.

Most of the characteristics mentioned in the introduction (sensitivities or hyporesponsiveness to various stimuli, responsiveness to loud sounds, dislike of water, hypersensitive taste, lack of attention, ignoring loud sounds and not responding to names) are common symptoms of autism spectrum disorder (ASD).

But the term “developmental disabilities” is much broader, encompassing diagnoses other than ASD.

My recommendation is to use the label “neurodevelopmental disabilities.”

According to the Diagnostic and Statistical Manual of Mental Disorders (DSM-5), this term includes children with ASD, intellectual disabilities, attention deficit hyperactivity disorder (ADHD), and various motor disorders (among others). These diagnoses are precisely those that appear in Table 1.

I think that the introduction should contain a definition of the term “neurodevelopmental disorders” and that this term should be used throughout the article.

Additionally, other symptoms should be added to the list of symptoms to characterize these children, such as cognitive limitations, difficulties in understanding social contexts or in identifying the causes of what is happening around them. These symptoms can be very relevant in natural emergencies.

I have changed “developmental disabilities” in the text to “neurodevelopmental disorders”. “CDD” has been changed to “CND”. In addition, the introduction and discussion have been revised.

Here are some other questions that I think the authors should consider:

  1. Introduction.

In the introduction practically no previous studies are mentioned that have addressed the same objectives as this article.

The introduction should be expanded by framing this article in the field of research on the location of shelters for natural emergencies for the general population; and for people with disabilities in particular.

P2 L60-83

Introduction

I added the following text:

For CNDs to evacuate quickly in the event of a disaster, it is necessary to know the location, route, and geographical characteristics of the nearest shelter and prepare for evacuation. Network analysis using geographic information systems (GISs) is suitable for determining the reachable range from the optimum shelter, route, and travel speed. Thus far, there have been few studies using GIS to help vulnerable people evacuate in the event of a tsunami. Analysis of shelter selection based on hurricane strength and shelter demand has been conducted using the interdiction and median model [16]. Evaluation of existing shelters using the p-median problem has been reported in a part of the target area (Aki City) of this study [17]. Estimations of human harm has been carried out using buffer analysis of a tsunami vertical shelter [18]. Evacuation time evaluation [19] is conducted to identify the population at risk of injury in a tsunami and considers the number of evacuation vehicles and congestion. Establishment of effective signposting to help the hearing impaired to evacuate [20] has been reported.

Reports on evacuation from a tsunami and evaluation of shelters have used sensitivity analysis with least-cost distance (LCD) modeling [21, 22], Evacuation routes and shelter allocation methods to minimize casualties [23] as well as evaluation of the effectiveness of protection in vertical evacuation from a near-field tsunami [24], have been reported. On reports targeting vulnerable people, Emergency Evacuation Readiness of Full-Time Wheelchair Users with Spinal Cord Injury [25], Simulation for vulnerable people to evacuate by car [26], Simulation of tsunami evacuation guidance signs for the hearing impaired [20], evacuation simulation of people using medical devices from a tsunami [27]. However, no studies have focused on local CNDs, estimated tsunami damage and evacuation times, or the geographic requirements to protect these vulnerable children.

  1. Emergency shelter for vulnerable people (ESVP).

This article may be of interest to researchers specialized in natural emergency response analysis who are very familiar with these issues.

But it may also be of interest to researchers interested in the analysis of public policies aimed at people with disabilities. In this case, these researchers will hardly have knowledge of natural emergencies and their response, and they need some context to understand the article.

Although in Japan it's probably be obvious, some issues should be explained in the introduction. For example:

Who decides where, when and with what features to build these ESVPs? The local governments, the national government?

Who bears the costs of its construction and maintenance?

Children with neurodevelopmental disorders who must live in these ESVPs for days, are they accompanied by their relatives? Or are they only accompanied by professionals?

This information should be added in lines 65-68.

I suggest authors to put themselves in the shoes of a person who is not at all familiar with earthquake or tsunami alerts, and to explain issues that, although they may seem basic in Japan, are not in many contexts.

P3 L97-105

I added a description about emergency shelters for vulnerable people (ESVPs):

Emergency shelter for vulnerable people (ESVP)

An ESVP is a facility specially prepared for people who have difficulty staying in general shelters, and their families. The Basic Law on Disaster Management defines an ESVP as a shelter for elderly people, pregnant women, people with illnesses and disabilities, and people with developmental disorders and intractable diseases. Kochi Prefecture has designated the existing welfare facility as an ESVP. The decision to open an ESVP in the event of a disaster is made by the head of the local government. If there are vulnerable people in a general shelter, the head of the local government will request the facility manager to open the ESVPs [7]. Most municipalities do not cover the costs of opening and maintaining ESVPs in the event of a disaster.

  1. Structure of the article.

3.1. Table 1.

Table 1 describes the characteristics of the participants. It should be relocated to the Materials and methods section, not the Results section.

P5 L139

I have moved Table 1 to the Materials and methods section.

3.2. Lines 200-209.

This content is more appropriate for the introduction section, not for the discussion.

I moved the following text to the introduction.

P2 L53-57

During the 1995 Great Hanshin-Awaji Earthquake, parents of CNDs often chose to keep them in cars or away from shelters to avoid disturbing others [12]. CNDs may exhibit problematic behaviors such as aggression, hyperactivity, and anxiety following unexpected changes in their schedule and stress owing to environmental changes [13, 14]. However, the symptoms of CNDs are difficult for others to understand. In the 2016 Kumamoto Earthquake, some CNDs were not allowed to stay in evacuation shelters because their unbalanced eating and panic were perceived as “selfish” [15].

3.3. Lines 244-254.

This content is repeated in the Conclusions section. It is suggested to delete these lines and leave their content only in the Conclusions section.

I deleted the text the text 1–6 in the discussion section.

  1. Figure 2.

Regarding the format of the figure 2, I consider that it is not necessary to represent this information with a graph. It would be enough to explain it through text.

Regarding its content, have these results been prepared from the results of the K-DiPS?

If so, it should be specified.

I deleted Figure 1 and added some text in the revised manuscript.

P5 L141-148

According to the K-DiPS checklist completed for the 47 CNDs included in the study, participants had the following characteristics: “Hyperactive and may go anywhere if left unattended”, 26 CNDs (55.3%); “Makes loud voices wherever you are owing to poor resistance to frustration”, 21 (44.7%); and”Extremely unbalanced diet”, 21 (44.7%); among others. The items regarding the need for help in the event of a disaster were as follows: “Hyperactive and may go anywhere if left unattended” 24 (51.1%) , “Makes loud voices wherever you are owing to poor resistance to frustration” 21 (44.7%), “Extremely unbalanced diet” 19 (40.4%) children.

  1. Lines 150-151.

Before referring to the specific shelters O and Q, the number of existing shelters should be reported and that each one is to be named by a letter.

It is possible that this is a well-known issue in Japan. Most of the potential readers of the article who do not live in areas threatened by tsunamis are not familiar with this.

I added the section “Overview of ESVPs” to the results.

P6 L194-199

4.1. Overview of ESVPs

In the target area, 17 facilities have been designated as ESVPs and have been assigned labels from A to Q. Of these, 5 (29.4%) ESPVs were located in the inundation prediction area; specifically, these were A, F, H, P, and Q. Information regarding inundation depth, straight-line distance from ESVP to the coast, CNDs expected to evacuate, and municipality is shown in Table 1.

  1. Discussion.

In the discussion, the results of the present study should be compared with those of other possible previous studies that have worked on this same issue.

P9 L270- 273

The effectiveness of vertical evacuation has been pointed out in evacuation from a near-field tsunami [24]. In Japan, it has been reported that tsunami evacuation towers for vertical evacuation may move from highlands to lowland tsunami evacuation towers, increasing the risk of damage [36]

P10 L284- 287

Generally, in areas at high risk of a tsunami, a strong, tall building is recommended as a vertical evacuation site [37, 38] however, instead of blindly heading to such a building, the location and tsunami flow in the immediate area should be considered, as well as the direction of arrival.

Formal issues:

  1. Keywords must be in alphabetic order.

P1 L24- 25

I changed the order of the keywords to alphabetical order.

  1. It must be indicated the meaning of “GIS” the first time it appears (it is currently explained on line 100).

P2 L62

I have defined “GIS” at its first mention in the text.

Round 2

Reviewer 1 Report

I am satisfied with the responses and therefore support the publication for this manuscript.